# Photoreceptors Regulate Plant Developmental Plasticity through Auxin

**DOI:** 10.3390/plants9080940

**Published:** 2020-07-24

**Authors:** Jesse J. Küpers, Lisa Oskam, Ronald Pierik

**Affiliations:** Plant Ecophysiology, Dept. Biology, Utrecht University, Padualaan 8, 3584 CH Utrecht, The Netherlands; j.j.kupers@uu.nl (J.J.K.); l.oskam@uu.nl (L.O.)

**Keywords:** auxin, shade avoidance, phototropism, polar auxin transport, developmental plasticity, auxin signalling, phytochrome, cryptochrome, phototropin

## Abstract

Light absorption by plants changes the composition of light inside vegetation. Blue (B) and red (R) light are used for photosynthesis whereas far-red (FR) and green light are reflected. A combination of UV-B, blue and R:FR-responsive photoreceptors collectively measures the light and temperature environment and adjusts plant development accordingly. This developmental plasticity to photoreceptor signals is largely regulated through the phytohormone auxin. The phytochrome, cryptochrome and UV Resistance Locus 8 (UVR8) photoreceptors are inactivated in shade and/or elevated temperature, which releases their repression of Phytochrome Interacting Factor (PIF) transcription factors. Active PIFs stimulate auxin synthesis and reinforce auxin signalling responses through direct interaction with Auxin Response Factors (ARFs). It was recently discovered that shade-induced hypocotyl elongation and petiole hyponasty depend on long-distance auxin transport towards target cells from the cotyledon and leaf tip, respectively. Other responses, such as phototropic bending, are regulated by auxin transport and signalling across only a few cell layers. In addition, photoreceptors can directly interact with components in the auxin signalling pathway, such as Auxin/Indole Acetic Acids (AUX/IAAs) and ARFs. Here we will discuss the complex interactions between photoreceptor and auxin signalling, addressing both mechanisms and consequences of these highly interconnected pathways.

## 1. Introduction

The phytohormone auxin is long known to steer plant growth and development to ensure optimal light capture. In 1880 Charles and Francis Darwin noticed and experimented on the phototropic bending of *Phalaris canariensis* coleoptiles towards a light source [1]. They observed that covering the coleoptile tip with tinfoil or a dark painted glass tube reduced their capacity to bend their coleoptile base, regardless of what light was locally perceived at the coleoptile base. Moreover, a tiny slit in the paint induced directed bending towards a weak light source even when the coleoptile base was brightly illuminated from another side. They concluded that “when seedlings are freely exposed to a lateral light some influence is transmitted from the upper to the lower part, causing the latter to bend”. Later experiments by Peter Boysen-Jensen, Frits Went and many others revealed that light-directed growth depends on the transport of a water-soluble chemical from the coleoptile tip towards the dark side of the coleoptile base. This chemical was named auxin [2]. Ever since, auxin has been studied intensively in plant biology and certainly also in relation to light cues. We now know how auxin is synthesized, (de)conjugated, transported, sensed and responded to. Here, we discuss recent updates on auxin-mediated growth responses to photoreceptor stimuli. We will discuss how light sensing intimately regulates all aspects of auxin biology, ranging from auxin synthesis from the amino acid tryptophan to transcription factors controlling the expression of auxin-responsive genes. We will also review the current understanding of how these photoreceptor-dependent modifications of auxin biology regulate the developmental plasticity needed for optimal performance under heterogeneous light conditions.

## 2. Photoreceptors to Sense Temperature and Light Quality, Quantity and Direction

Light capture is essential for plant development as it fuels photosynthesis. Therefore, plants display elaborate plasticity to fine-tune their growth to the prevailing light conditions. Elongation and phototropic bending of the hypocotyl towards light, as well as upward leaf movement and petiole elongation enhance access to light in dense vegetation and are thus adaptive shade avoidance growth responses [3]. Plant leaves absorb blue (B) and red (R) light for photosynthesis whilst reflecting far-red (FR), resulting in a low ratio of R to FR light (R:FR) and low B light in vegetational shade [4,5]. These light composition changes are carefully monitored by several classes of wavelength-specific photoreceptors [3]. Among those are R:FR perceiving phytochromes, blue light sensitive cryptochromes and phototropins and UV-B responsive UV Resistance Locus 8 (UVR8). The extent to which adaptive photoreceptor-driven growth responses are induced depends on the combined light-induced photoreceptor activation and their shared control of auxin signalling (Figure 1).

### 2.1. Phytochromes

There are five phytochromes (phyA-E) in *Arabidopsis*. The growth response to low R:FR mainly occurs through inactivation of phyB, with minor additional function for phyD and phyE (reviewed in [6]). PhyB is synthesized as inactive Pr and photoconverted to active Pfr by R light. Darkness and relative high abundance of FR will cause a reversion back to Pr. PhyB photoconversion results in a ratio of Pfr/Pr that resembles the R:FR [6]. Besides the role of phyB in light signalling, thermal acceleration of the conversion of Pfr to Pr stimulates the growth response to high ambient temperature [7,8], which enhances leaf cooling [56]. In contrast to phyB, phyA is active in very low R:FR and prevents excessive growth in such unfavourable conditions [57]. Active phyB translocates to the nucleus where it interacts with and inactivates several Phytochrome Interacting Factor (PIF) bHLH transcription factors. PhyB inactivates PIF1, PIF3, PIF4 and PIF5 through phosphorylation, ubiquitination and degradation, while PIF7 gets phosphorylated and inactivated without rapid degradation (reviewed in [9]). Inactivation of phyB releases PIF repression, resulting in enhanced target gene expression, auxin synthesis and growth.

### 2.2. Cryptochromes

Next to the decreased R:FR, the fluence rate of blue light is also decreased in shade. Perception of low B by the flavoproteins cryptochrome 1 (cry1) and cry2 results in hypocotyl elongation and enhanced low R:FR-mediated petiole elongation in *Arabidopsis* [55,58,59,60]. In blue light, cry1 and cry2 interact with PIF4 and PIF5 and repress their transcriptional activity [55,61]. While PIF7 plays a major role in low R:FR-mediated hypocotyl elongation through stimulating auxin synthesis, with additional roles for PIF4 and PIF5, it is less strictly required for hypocotyl elongation in low B [55,60]. However, cryptochrome inactivation has recently been shown to affect hypocotyl phototropism towards blue light via PIF4, PIF5 and PIF7 and YUCCA-mediated auxin synthesis [10].

### 2.3. Phototropins

Phototropism, or bending towards a light source, primarily depends on blue light-induced autophosphorylation of the plasma membrane-associated AGCVIII kinases phototropin 1 (phot1) and phot2 [41]. Whilst phot1 regulates bending in a wide range of blue light intensities, phot2 primarily functions in high blue light. Unilateral blue light perception by phot1 leads to dephosphorylation of the key interacting protein Non-Phototropic Hypocotyl 3 (NPH3) [41]. Dephosphorylated NPH3 dissociates from the phot1 membrane complex to form cytosolic aggregates which occur most strongly in the illuminated side of the hypocotyl [42]. This NPH3 dephosphorylation asymmetry is mirrored by auxin reporters [10,43,44,46]. Asymmetric auxin concentrations allow for phototropic hypocotyl bending through auxin-mediated cell elongation on the shaded side of the hypocotyl [62]. Although most research regarding phototropism has focused on hypocotyl bending, phototropin-mediated bending towards blue light also occurs in inflorescence stems and petioles [63]. Furthermore, besides bending to blue light, inflorescences and hypocotyls also bend towards unilateral UV-B using the UV-B receptor UVR8 [64,65] (reviewed in [66]). 

### 2.4. UVR8

Upon UV-B irradiation, the inactive UVR8 dimer monomerizes and relocates to the nucleus where it stabilizes the growth repressing transcription factor Elongated Hypocotyl 5 (HY5) (reviewed in [67]). This HY5 accumulation reduces gibberellic acid (GA) signalling by promoting GA inactivation through increased *GA2OX1* transcription [11]. Moreover, as UV-B sensing by UVR8 serves as a signal for full sunlight, UVR8 signalling inhibits the elongation responses to low R:FR and high temperature through degradation of PIF4 and PIF5 [11,68,69,70]. Moreover, UVR8 signalling stabilizes Long Hypocotyl In Far-Red 1 (HFR1), which negatively regulates PIF4 and PIF5 activity [19]. The combined inactivation of PIFs and accumulation of HY5 reduces auxin and gibberellin signalling and thereby dampens the growth response to low R:FR and elevated temperature.

## 3. Photoreceptor Control of Auxin Synthesis and Conjugation

Photoreceptor signalling regulates local and systemic auxin concentrations at three levels of regulation: biosynthesis, (de)conjugation and transport (Figure 1) (reviewed in [71]). Bioactive auxin, indole-3-acetic acid (IAA), is mainly synthesised in a two-step pathway from its tryptophan (Trp) precursor [20,21,22,23,24]. Trp is converted to indole-pyruvic acid (IPyA) by Tryptophan Aminotransferase of *Arabidopsis* 1 (TAA1) and TAA1-Related proteins (TARs) [21,23,24]. IPyA is next converted to IAA in a rate limiting step by YUCCA (YUC) flavin monooxygenases [20,22,24]. Although a *taa1* mutant was also characterised in a shade avoidance mutant screen as *shade avoidance 3* (*sav3*), *TAA1* and *TAR* expression are not typically stimulated by low R:FR [23]. On the other hand, low R:FR and high ambient temperature do stimulate *YUCCA* gene expression through PIF4, PIF5 and PIF7 [11,13,14,15,16,17,18]. Indeed, low R:FR and elevated temperature promote IAA accumulation in the shoot [13,14,15,23], and even in elongating hypocotyls specifically [45,72].

The conversion of IPyA to IAA is described to be rate-limiting in IAA synthesis. Reduced IPyA levels inhibit IAA synthesis and auxin-mediated hypocotyl elongation in shade and elevated temperature [25,26]. IPyA can be reverted to Trp by the aminotransferase Reversal of SAV3 Phenotype 1 (VAS1), which inhibits shade-induced hypocotyl elongation [25]. Moreover, IPyA glycosylation via UDP-Glycosyltransferase 76F1 (UGT76F1) also reduces hypocotyl elongation [26]. PIF4 stabilisation in elevated temperature reduces *UGT76F1* expression leading to lower levels of glycosylated IPyA, which ultimately increases IAA levels in elevated temperature [26].

Although seedlings elongate their hypocotyls in low R:FR, it has been shown in *Brassica rapa* seedlings that low R:FR triggers auxin synthesis in the cotyledons, which, indeed, are the classic sites of auxin synthesis [72]. Auxin then is transported to the hypocotyl to promote elongation (Figure 2) [45]. Consistently, transcriptome surveys in *Arabidopsis* indicate light sensing and auxin synthesis in the cotyledons, whereas mostly downstream responses are observed in the hypocotyl [73,74]. Another example of spatial separation between the location of light sensing and growth response can be observed in adult *Arabidopsis* plants. Low R:FR light perception at the leaf tip locally triggers auxin synthesis through PIF7-mediated upregulation of *YUC8* and *YUC9* expression [75]. The newly synthesized auxin is subsequently transported towards the petiole base, where petiole hyponasty is induced (Figure 2) [75,76].

Auxin synthesis upon phytochrome inactivation in seedlings mainly takes place in the cotyledons. However, auxin levels in the elongating hypocotyl may also be locally regulated by altered auxin conjugation and inactivation. Amino acid conjugation to auxin is mediated by clade II members of the Gretchen Hagen 3 (GH3) family of acyl acid-amido synthetases [27]. These GH3s reduce free IAA levels by IAA conjugation with different amino acids [27]. Because of functional redundancy, only higher order mutants display increased IAA concentrations [27,78]. In a mutant screen for suppressors of the R:FR-irresponsive *sav3* phenotype, Reverse of SAV3 Phenotype 2 (VAS2) was identified and shown to be GH3.17 [79]. GH3.17 conjugates IAA to glutamic acid (Glu) which is irreversible and leads to IAA degradation. The expression of *GH3.17* is only mildly reduced by shade in the hypocotyl [79], asking the question if this is a major node of regulation during shade avoidance. Perhaps a concerted downregulation of multiple GH3’s would indicate a major point of regulation. Importantly, the *vas2*/*gh3.17* mutant could elongate its hypocotyl in shade even without auxin transport from the cotyledons [79], indicating the potential for modulation of auxin concentrations in physiologically meaningful ranges without de novo synthesis. 

## 4. Photoreceptor Control of Auxin Transport

As mentioned earlier, cotyledon- and leaf-generated auxin has to be transported to specific target cells in order to initiate shade avoidance responses. Importantly, polar auxin transport (PAT) is required to relay photoreceptor information through the plant (reviewed in [80]). In PAT, auxin can enter the cell via passive influx as protonated auxin [80], or through Auxin1/Like AUX1 (AUX/LAX) auxin influx permeases (reviewed in [81]), and is directed to the neighbouring cell via polarly localised PIN-Formed 1 (PIN1) -PIN4 and PIN7 efflux carriers [82,83]. Polar auxin transport through PINs is essential for photoreceptor-mediated growth responses including hypocotyl elongation, phototropic bending and petiole hyponasty. Moreover, the ATP-Binding Casette Transporters of the B Subfamily (ABCB) auxin transporters are also involved in the regulation of auxin transport during photomorphogenesis and phototropic bending [84,85,86]. 

Auxin transport through PIN proteins is essential for low B and low R:FR mediated hypocotyl elongation [45,59]. In seedlings, low R:FR leads to redistribution of PIN3 in the hypocotyl endodermis from a downward apical orientation towards a more lateral outward orientation (Figure 2) [45]. This redirects the cotyledon-generated downward auxin flow towards the hypocotyl epidermis where cell elongation allows for elongation of the whole organ [45,87]. This lateral PIN3 redistribution occurs similarly on all sides of the hypocotyl, leading to uniform and upward growth. In contrast, phototropism and petiole hyponasty are the result of differential growth between two sides of the responsive organ. 

Hypocotyl phototropism towards unilateral blue light is the result of enhanced auxin signalling on the shaded side of the hypocotyl [10,43,44,46,88]. The observed auxin asymmetry is reduced in *pin3* and *phot1* mutants [43], and PIN1, PIN3, PIN4 and PIN7 are all required for normal phototropic bending [47,89]. Unilateral light triggers clathrin-mediated internalisation of PIN3 from the outer endodermal membrane in the illuminated side (Figure 1) [43,44]. The ensuing asymmetric PIN3 localisation redirects the auxin flow towards the shaded side, stimulating growth towards the light by enhanced cell elongation on the shaded side (Figure 2).

Phototropic hypocotyl bending is not limited to dark-grown seedlings. In fact, de-etiolation by blue or red light renders the seedling more responsive to subsequent unilateral blue light, possibly through inactivation of PIFs by cry and phy [42]. Moreover, phyB and cry1 inactivation in shade stimulate phot1-mediated hypocotyl bending towards blue light when compared to white light conditions [10,12]. This regulation requires auxin synthesis via PIF4, PIF5 and PIF7-dependent regulation of YUCCAs [10,12]. In this model, the increased auxin flow into the hypocotyl from the cotyledons [45,72] can feed into a phot1-mediated PIN3 asymmetry and increase auxin concentrations on the shaded side.

In addition to phototropin-mediated blue light signalling, UV-B signalling via UVR8 can also regulate phototropic bending [64,65]. Although unilateral UV-B does result in an auxin signalling gradient in hypocotyls, this seems to depend less strictly on polar auxin transport than does blue light-dependent phototropism [65]. A UVR8-dependent HY5 gradient was observed between the UV-B-illuminated and non-illuminated side of the hypocotyl [90] and this could differentially affect auxin response between these two sides [65], although further experiments are needed to resolve this. Similar mechanisms may regulate inflorescence bending towards UV-B light, where the observed HY5 gradient is mirrored by auxin and GA signalling gradients [64].

During hyponastic leaf bending, PIN-mediated auxin transport from the leaf tip towards the petiole base is essential [75,76]. Indeed, blocking auxin transport from the leaf tip using local application of the polar auxin transport inhibitor NPA, strongly impaired this response [75]. Although PIN3 is a major regulator in hyponastic leaf movement, it acts redundantly with PIN4 and PIN7 in this response [75,76]. So far, it is unknown if and how an abaxial/adaxial auxin response gradient builds up in the petiole base itself in response to auxin coming from the leaf tip under low R:FR conditions. However, this has been studied in the hyponastic growth response to elevated ambient temperature [77]. In such conditions, PIN3 was shown to concentrate on the outer side of the abaxial endodermis [77], suggesting directed auxin flow towards the elongating, abaxial side of the petiole (Figure 2). PIN3 accumulation on the abaxial endodermis requires functional PIF4 and Asymmetric Leaves 1 (AS1), an important regulator of abaxial/adaxial cell identity [77]. Removal of the lamina prior to high temperature exposure abolished petiole hyponasty, suggesting that auxin transport from the lamina to the petiole may be required for temperature-induced petiole hyponasty [77]. More subtle manipulations, for example locally using the polar auxin transport inhibitor NPA, would help ascertain that this is really the case. 

Besides PIN-mediated auxin transport, regulated diffusion through plasmodesmata may stimulate rapid directional auxin transport from leaf tip to petiole base [91]. The direction of the auxin flow depends on higher permeability for auxin in the longitudinal versus transverse direction of the cells along the petiole and midrib of the lamina [91]. Impaired glucan-mediated control of plasmodesmata aperture in the Glucan Synthase Like 8 (GSL8) mutant *gsl8* reduced the leaf hyponasty response to auxin application to the leaf tip [91]. It remains to be investigated if plasmodesmata aperture is regulated by photoreceptor signalling and light or temperature treatments.

## 5. Regulation of PIN Relocalisation by Photoreceptor Signalling

Although our understanding of photoreceptor effects on PIN localisation is quite extensive, the exact mechanisms remain uncertain. PIN polarization is regulated by subcellular trafficking of PINs, which is a constant process that is influenced by light, but also other environmental stimuli such as gravity, temperature and salinity (reviewed in [80]). Regulation of PIN localisation, as well as activation via phosphorylation, occurs through different components such as ARF-GEF GNOM and three families of protein kinases, AGC kinases, Mitogen Activated Protein (MAP) Kinases (MPKs) and Ca^2+^/calmodulin-dependent protein kinase-related kinases (CRKs) [80,92]. Several members of the AGCVIII kinase family, to which phototropin also belongs, have been implied to regulate auxin transport and PIN phosphorylation during phototropism [43,46,47]. Transcript levels of one of these AGCVIII kinases, *PINOID* (*PID*), were found to be reduced by light [43]. This was linked to reduced PIN3 asymmetry in seedlings mis-expressing *PID* [43]. However, the quadruple *pid pid2 wag1 wag2* mutant only has reduced phototropism in R light pre-treatment conditions but bends normally in other treatments [93]. Petiole hyponasty, induced by elevated temperature, was found to be correlated with PIF4-induced *PINOID* (*PID*) expression in the elongating abaxial side of the petiole, where auxin is thought to accumulate [77]. In such conditions, ectopic *35S*::*PID* expression disturbed PIN3 localisation and inhibited temperature-mediated leaf hyponasty [77].

Other AGCVIII kinases that are required for hypocotyl bending include D6 Protein Kinase (D6PK), D6PK Like 1 (D6PKL1), D6PKL2 and D6PKL3 as well as AGC1-12 [46,47]. Similar to PID, both D6PKs and AGC1-12 are able to phosphorylate the PIN1 hydrophilic loop in vitro [46]. However, D6PKs and AGC1-12 phosphorylate at least one unique PIN1 serine residue that is not phosphorylated by PID, which might distinguish them from PID in regulating hypocotyl bending [46]. The function of D6PKs and AGC1-12 in regulating hypocotyl bending appears to extend beyond phototropism as their mutants also show decreased gravitropism [46]. Moreover, expression of *D6PK* and *D6PKL1* is specifically induced by low R:FR in hypocotyls and *d6pk01* mutant and *D6PK* overexpressing seedlings display reduced hypocotyl elongation in low R:FR [73].

## 6. Phytochrome Signalling Regulates Auxin Perception

Photoreceptors not only influence auxin concentrations, but also regulate downstream auxin perception and signalling. Auxin perception mainly occurs via the nuclear Transport Inhibitor Resistant1 (TIR1) and Auxin Signalling F-Box (AFB) receptors of the SKP-Cullin-F-Box (SCF)^TIR1/AFB^ ubiquitin ligase complex that form a receptor complex with their Auxin/Indole Acetic Acid (AUX/IAA) coreceptors (Figure 1) [28] although other mechanisms of auxin perception have also been implied (reviewed in [94]). In persistent low R:FR, TIR1 and AFB2 are required for hypocotyl elongation and their transcript levels are increased [32]. Increased *TIR1* and *AFB2* transcripts coincide with reduced *microRNA393 (miR393)* expression. *miR393* targets *TIR1*, *AFB2* and *AFB3* transcripts, reducing auxin signalling in adverse environmental conditions (reviewed in [95]). Reduced *miR393* expression in persistent low R:FR suggests enhanced auxin activity, consistent with higher activity of the auxin reporter DR5::GUS in *mir393a mir393b* double mutant seedlings in persistent low R:FR [32].

Once the SCF^TIR1/AFB^ complex binds auxin, it interacts with and ubiquitinates the auxin signalling repressors AUX/IAAs, which are subsequently degraded [28]. This releases AUX/IAA repression of auxin related gene expression and cell growth [28]. Recent studies have established that photoreceptor activation by light prevents AUX/IAA degradation and thereby lessens the auxin induced growth response [29,30,31].

## 7. Photoreceptor-Mediated AUX/IAA Stabilisation Reduces ARF Activity 

Cry1 reduces hypocotyl growth and DR5::GUS auxin reporter activity through blue light intensity-dependent stabilisation of AUX/IAAs (Figure 1) [29]. Activated cry1 binds AUX/IAAs and thereby reduces their interaction with TIR1 [29]. Comparable to cry1, phyB can interact with and stabilise AUX/IAAs with increasing R light intensity [29]. 

Unlike phyB, phyA is stabilised in deep shade and represses excessive auxin-induced gene expression and hypocotyl elongation in such unfavourable conditions [30,57]. Just like cry1 and phyB, phyA can interact with AUX/IAAs in the nucleus, tentatively preventing TIR1-dependent degradation [30]. AUX/IAA protein levels are reduced in mild shade, presumably due to reduced phyB activity, but in deep shade strong phyA activity outcompetes TIR1 for AUX/IAA binding [30].

AUX/IAAs repress auxin signalling through inhibition of the Auxin Response Factor (ARF) transcription factor-mediated gene expression [95]. The transcription activating class A ARFs consisting of ARF5, ARF6, ARF7, ARF8 and ARF19, constitute the main AUX/IAA targets [95]. Of these class A ARFs, ARF6, ARF7 and ARF8 are redundantly required for auxin mediated hypocotyl elongation in low R:FR and high temperature [40]. In a recent study, ARF6 and ARF8 were found to interact with phyB in R light and cry1 in blue light, resulting in reduced ARF6/ARF8 DNA binding [31]. In correspondence with the observed light-induced stabilisation of AUX/IAAs by phyB and cry1 [29], phyB and cry1 were shown to stimulate AUX/IAA-ARF interaction and AUX/IAAs were shown to strengthen the photoreceptor mediated inhibition of ARF DNA binding (Figure 1) [31]. In shade, the combination of increased IAA concentration and reduced light would disrupt the growth-repressive photoreceptor-AUX/IAA-ARF complex, thereby allowing for ARF-mediated gene expression and hypocotyl elongation.

## 8. Photoreceptor Control of the BAP/D Module 

In shade and elevated temperature, auxin concentrations increase rapidly. However, this increase is often transient and lost on the second day of treatment [32,96,97]. A subsequent increase of auxin sensitivity is required to maintain auxin signalling for a longer duration in low R:FR [32]. Moreover, other hormones may further stimulate growth beyond the first day [96]. The synthesis of gibberellic acid (GA) is increased in low R:FR through enhanced *GA20-OXIDASE* transcription [96,98,99]. Increasing GA concentrations promote degradation of growth-repressive DELLA proteins. DELLAs are nuclear localised repressors that inhibit the activity of many transcription factors including the BR responsive growth promotors Brassinazole Resistant 1 (BZR1) and its close homolog BRI1-EMS-Suppressor 1 (BES1) as well as ARFs and PIFs [33,34,36,100]. These transcription factors together with their DELLA repressor constitute the BZR-ARF-PIF/DELLA (BAP/D) module (Figure 1) [36]. Similar to PIF and ARF, the third TF of this group, BZR1, is also inactivated upon light-activation of phyB and cry [37,38,39]. 

BZR1, ARF6 and PIF4 stimulate cell growth through induction of many shared target genes, and they interact and reinforce each other’s activity at those targets [35,36]. These interactions would explain why PIFs and BZR stimulate auxin sensitivity [14,36]. However, each member of the BAP module also has its own specific targets [36], as illustrated by the observation that *arf6 arf7 arf8*, although not responsive to exogenous IAA, maintains the hypocotyl elongation response to exogenous BR and GA treatment [40]. All taken together, this implies a complex growth promoting network of interacting transcription factors that are stimulated by auxin, gibberellin and brassinosteroid signalling, whilst being repressed by active phy and cry photoreceptors.

## 9. Auxin-Modulated Cell Growth

The auxin-mediated cell elongation response to unilateral light, shade, neighbour proximity signals and high temperature depends on enhanced expression of *SAUR19-24*, members of the *SMALL AUXIN UP RNA* family [13,62,73,101]. In the *Arabidopsis* hypocotyl *SAUR19* expression is limited to the epidermis which is the cell layer that restricts hypocotyl growth [87], and in unilateral blue light *SAUR19* expression only occurs on the shaded side of the hypocotyl [62]. Activation of SAUR19 stimulates H^+^-ATPase proton pumps, which leads to rapid apoplast acidification and acid growth (Figure 1) [48,49,50]. In shade-exposed *Arabidopsis* petioles, apoplast acidification happens within minutes [52]. The acidification is accompanied by enhanced expression and activity of cell wall-modifying proteins, such as Xyloglucan Endotransglucosylase/Hydrolases (XTHs) and expansins [50,51,52]. At least part of the *XTH* induction in *Arabidopsis* is auxin-dependent [53], but PIFs can also directly regulate *XTH* expression [54,55].

## 10. Preventing Excessive Growth

We described before how cryptochrome and phytochrome inactivation both stabilise BZR/ARF/PIF proteins and increase auxin and gibberellin levels. In such conditions, the negative regulators AUX/IAA and DELLA are removed. Furthermore, auxin sensitivity increases in persistent shade. It would, therefore, seem pertinent for the plant to employ precise and dedicated negative feedback to prevent excessive growth.

This is achieved in deep shade by activation of phyA which leads to stabilisation of AUX/IAAs [30]. In addition, several of the most strongly upregulated transcripts in shade include negative regulators of auxin and shade signalling. Such transcripts include *AUX/IAA*s, *GH3*s, *PIL1*, *HFR1*, *PAR1* and *PAR2* [102,103] which are also frequently used as marker genes for auxin response and photoreceptor inactivation. PIL1, HFR1 and PARs are bHLH proteins that can physically interact with PIFs, reducing PIF binding to target gene promoters, including auxin-associated genes (reviewed in [103]). Enhanced *AUX*/*IAA* expression and protein accumulation would reduce the auxin response by reducing ARF transcriptional activity [95]. However, in specific conditions AUX/IAAs may indirectly stimulate the auxin response. PIF4 was shown to increase the expression of *IAA19* and *IAA29* in persistent shade [32]. The expression of these *AUX*/*IAA*s appears to stimulate hypocotyl growth, possibly through inhibiting auxin-induced expression of the stronger negative growth regulator *IAA17* [32]. It would be interesting to further tease apart the interaction between AUX/IAAs in variable conditions.

## 11. Future Perspectives

We have reviewed here how auxin is a central node of photoreceptor-dependent regulation of plant development plasticity. Multiple interactions have been identified between photoreceptor activity and auxin biology, spanning all levels ranging from auxin synthesis, to response. Although it may seem as if most of photoreceptor-driven auxin biology is understood, much of this is still relatively early days. We will outline a few future perspectives for this field in the coming years, but this is by no means complete.

Auxin biology itself is still only partly understood. The massive diversity in auxin synthesis, transport and response factors makes it difficult to understand the complete interactome of photoreceptors with auxin. Even if we know that, for example, ARF6 and ARF8 interact with PIF4, there are countless other ARFs (and PIFs) for which this still needs to be resolved. The high redundancy as well as diversification within these protein families helps plants to respond to various and subtle changes in environmental and developmental signals. The many possible interactions of these auxin-associated proteins with photoreceptor signalling enable subtleties in developmental plasticity that we may not appreciate to their full potential yet. Resolving the multiple possible points of crosstalk will undoubtedly unravel novel subtleties in developmental plasticity.

In order to address the full potential of auxin-driven developmental plasticity to light, a challenge in future research will be to pair relatively simple study systems on etiolated seedlings exposed to monochromatic light, with light-grown adult plants and the full multi-colour perspective of natural daylight. With the exception of the studies of phyA-AUX/IAA interaction [30], the work on photoreceptor regulation of AUX/IAAs and ARFs has for example been done in monochromatic light. It would be interesting to see if phyB-ARF-AUX/IAA binding is reduced with decreasing R:FR in white light backgrounds and if cry1-ARF-AUX/IAA binding is reduced with reducing blue light intensity. Furthermore, given the fluctuating levels of blue, red and far-red inside natural vegetations, these photoreceptors may even interactively regulate ARF and AUX/IAA activity. Moreover, the observed interactions have mainly been studied in seedlings. The ease of use of the seedling system fully justifies the chosen method. However, experimentation in adult plants can provide more detailed insights in the spatiotemporal activity of these regulatory mechanisms [75,76,104].

Finally, much is to be resolved about whole plant coordination of spatially distinct light signals. For example, one leaf may be exposed to other light cues than another leaf [105]. Given its mobility, and the tight control over this, auxin transport would be an obvious candidate mediator of such integration. Indeed, PIN proteins are known to facilitate auxin transport from site of light detection to the site of action [75,76]. Analogous to the root tip, where different PINs have unique localisation domains and mediate auxin transport in specific directions [106], it will be helpful to extend our understanding of the localisation domains, and their plasticity in response to different photoreceptor cues, of the different PINs in the different shoot organs. Photoreceptor-regulated transcription of the PIN-phosphorylating AGCVIII kinases *PID* and *D6PK* has been reported in some conditions [43,73,77]. Moreover, misexpression of *PID*, but not *D6PK*, has been shown to alter PIN3 localisation [43,77,107]. It will be interesting to see how our understanding of AGCVIII kinase-mediated PIN regulation by photoreceptor signalling develops. Generating new mutant alleles and higher order mutants and testing those in various conditions might considerably deepen our understanding of photoreceptor-controlled auxin transport.


**Outstanding Questions**
1. What is the full network of auxin signalling, transport and response interactions during photoreceptor signalling?2. How do natural combinations of light cues control the complexity of auxin-driven developmental plasticity?3. Which spatiotemporal localisations of auxin transporters translate heterogeneous light cues to spatially explicit growth responses at the whole plant level?

## Figures and Tables

**Figure 1 plants-09-00940-f001:**
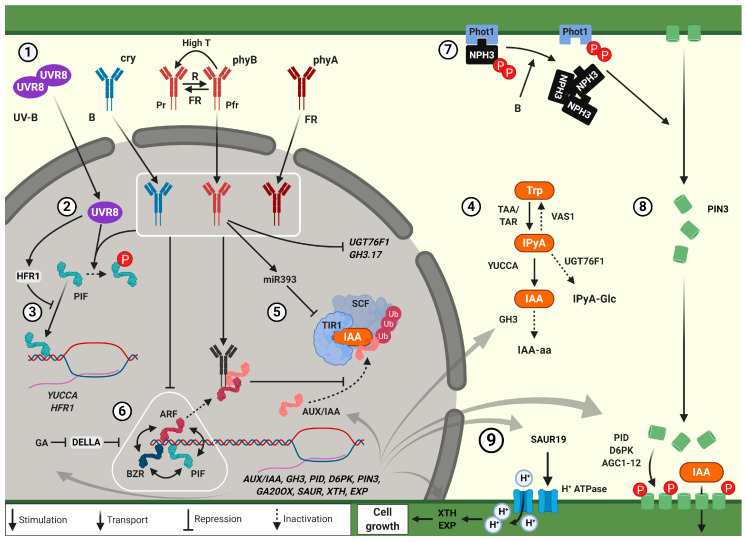
Photoreceptors regulate cell growth through altered auxin synthesis, transport and signalling. **➀** Wavelength-specific activation of the photoreceptors UV Resistance Locus 8 (UVR8), cry, phyB and phyA triggers their nuclear accumulation [3]. PhyB activation by red light is reversed by far-red light and spontaneous conversion that is accelerated at high temperature [6,7,8]. **➁** Active photoreceptors trigger Phytochrome Interacting Factor (PIF) phosphorylation, which leads to degradation for PIF4 and PIF5 and inactivation for PIF7 [9]. **➂** Free PIFs bind to promotors of *YUCCAs*, *HFR1* and many other target genes and stimulate their expression [10,11,12,13,14,15,16,17,18]. HFR1, which is stabilised in UV-B via UVR8, inhibits DNA binding of PIFs [19]. **➃** Auxin synthesis mainly occurs in a two-step pathway [20,21,22,23,24]. Trp is first converted to IPyA by TAA1 and TARs [21,23,24]. IPyA is next converted to active IAA auxin via YUCCA [20,22,24]. Negative feedback on IPyA levels occurs through reversal to Trp via VAS1 and IPyA glucosylation by UGT76F1 [25,26]. IAA is also inactivated by conjugation to amino acids via GH3 proteins [27]. **➄** In the nucleus, IAA interacts with the TIR1/AFB receptors of the SCF^TIR1/AFB^ receptor complex. Upon IAA binding, SCF^TIR1/AFB^ ubiquitinates Auxin/Indole Acetic Acids (AUX/IAA) proteins, which leads to AUX/IAA degradation [28]. In the absence of IAA, AUX/IAAs inhibit auxin signalling by interacting with Auxin Response Factors (ARFs), preventing their DNA binding and transcriptional activity. ARF activity is further reduced by photoreceptor stabilisation of AUX/IAAs, and the formation of a transcriptionally inactive photoreceptor-AUX/IAA-ARF complex [29,30,31]. PhyB inactivation in persistent shade enhances auxin signalling through reduced expression of the *TIR1*-targeting miR393 [32]. **➅** The transcriptional activity of ARFs is reinforced by the formation of a trans-activating transcription factor module together with BZR and PIF [33,34,35]. BZR1, ARF and PIF are all inhibited by interaction with growth-repressive DELLA proteins, forming the BAP/D module [36]. DELLA repression is alleviated by GA-mediated DELLA degradation in persistent shade conditions. Besides DELLAs, various active photoreceptors have also been shown to inhibit the activity of BZR1, ARF and PIF [37,38,39]. Active BZR1, ARF and PIF target many shared and unique target genes, including genes involved in auxin inactivation and transport, as well as gibberellin synthesis and cell growth [36,40]. **➆** Phot1 associates with NPH3 at the plasma membrane [41]. Phot1 activation by unilateral blue light leads to phot1 autophosphorylation. This triggers NPH3 dephosphorylation and a loss of PIN3 from the outer endodermal plasma membrane on the illuminated side of the hypocotyl (for details see Figure 2) [42,43,44]. **➇** Polar redistribution of PIN3 occurs in response to photoreceptor cues [43,44,45]. Moreover, PIN3 can be phosphorylated by PID, D6PK and AGC1-12 kinases that are required for various photoreceptor-mediated growth responses [43,46,47]. Polar localisation of PIN3 allows for directed auxin flow towards target tissues (for details see Figure 2). **➈** Auxin stimulates apoplast acidification through SAUR19-mediated activation of H^+^-ATPases [48,49,50]. This enhances the activity of cell wall modifying enzymes and results in acid growth [51,52,53,54,55]. This figure was created using BioRender.com.

**Figure 2 plants-09-00940-f002:**
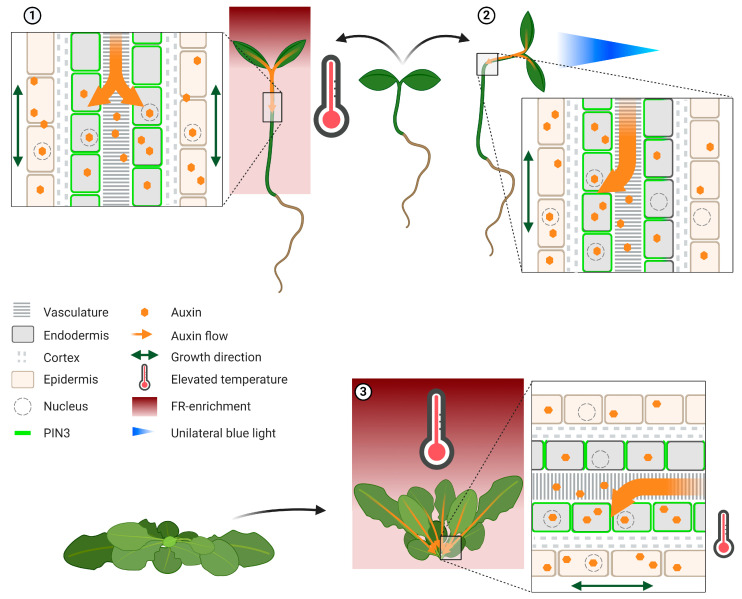
Photoreceptor control of auxin distribution patterns. Phenotypic changes of seedlings and adult plant after photoreceptor (de)activation are shown relative to plants grown under control conditions. Inserts depict a cellular representation of auxin localisation in either hypocotyl or petiole. **➀** Hypocotyl elongation of seedlings after photoreceptor phyB inactivation by elevated temperature and FR-enriched light. De novo synthesized auxin is transported from the cotyledons towards the hypocotyl, where an even distribution of PIN3 proteins between the different sides of the hypocotyl endodermis facilitates both downward and lateral auxin transport, allowing auxin accumulation throughout the hypocotyl [45,72]. This is different from the control white light situation where PIN3 localisation mostly facilitates rootward auxin transport. **➁** Unilateral blue light (here from the right) results in phot-dependent phototropic bending towards the light source. This occurs through asymmetric localisation of PIN3 proteins in the endodermis, favouring auxin transport towards the non-illuminated side [43,44]. The resulting auxin gradient promotes differential cell elongation that results in bending towards the light. **➂** Adult plant exposure to supplemental FR or elevated temperature results in upward leaf movement. PIN3 localisation towards the abaxial sides of the abaxial endodermal layer was observed at elevated temperatures, and this would lead to auxin accumulation on the abaxial side of the petiole [77]. In both elevated temperature and FR-enriched light, auxin and polar auxin transport are required for the hyponastic leaf movement [75,76,77]. This figure was created using BioRender.com.

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
