# Peer review of "Photoreceptors Regulate Plant Developmental Plasticity through Auxin"

_plants, 2020, doi:10.3390/plants9080940_

Round 1

Reviewer 1 Report

In this review by Küpers et al, the authors provide a comprehensive overview of how light cues are integrated into plant development and plasticity through the modulation of the hormone auxin. The review is well written and provides a good overview of the field. The figures accompanying the review are also very informative and summarize well the detailed text. I would find it helpful to have a 3rd figure summarizing the outstanding questions and future directions, but I do not think this is strictly necessary.

The only point that I think must be addressed is the omission of PGP-type auxin transporters as regulators of photomorphogenesis in Arabidopsis (see for example Nagashima et al, 2008, Plant J and Lin & Wang, 2005, Plant Phys, among others).

Author Response

We thank the reviewer for their suggestion to highlight the main future perspectives. Hence, we have added a short textbox highlighting outstanding questions and future directions

  As for PGP's, we have inserted a sentence (lines 223-225) about the involvement of PGPs (most commonly named ABCBs) in auxin transport during photomorphogenesis and phototropic bending, citing the indicated references.

Reviewer 2 Report

The manuscript presented by Küpers et al. focuses on the molecular mechanisms, required to link the photoreceptors’ light perception to growth regulation by auxin. Many laboratories investigate the fine details of these processes, thus it is worth to write such a review summarizing the major findings and orientate the readers towards the original studies. The manuscript is well-written, easy to follow and I found only minor issues which I recommend to deal with before publication.

Line 67: Besides triggered by FR light, Pfr to Pr conformational change does not only happen in high temperature. This spontaneous conversion is an intrinsic property of the PHYB receptor, which happen at any temperature (with different speed).

Figure 1:

- I propose to include the corresponding references to the statements in the figure legend. This would help the readers to explore further each specific step. The references are indicated by numbers; thus such an addition would not disturb the original structure of the text.

- I think, it may confuse the readers that whereas the PHOT directed processes occur in the membrane and cytosol, are clearly linked to cell growth; the molecular actions happen in the nucleus are not. I feel, that only a few arrows and some sentences explaining how nuclear signalling reaches the cell wall and affects growth are missing.

Optionally a paragraph could be inserted, discussing shortly what happens in darkness when photoreceptor mediated pathways are inactive explaining how auxin pathways contribute the skotomorphogenic growth. Maybe a modified Fig 1. could demonstrate how auxin signalling is working without photoreceptor action.

Author Response

Line 67: Besides triggered by FR light, Pfr to Pr conformational change does not only happen in high temperature. This spontaneous conversion is an intrinsic property of the PHYB receptor, which happen at any temperature (with different speed).

>>Very true, we have now elaborated this  (line 69 in current version of the paper), as well as line 106 to be more precise.

Figure 1:

- I propose to include the corresponding references to the statements in the figure legend. This would help the readers to explore further each specific step. The references are indicated by numbers; thus such an addition would not disturb the original structure of the text.

>>We thank the reviewer for this suggestion and added reference numbers to the figure legend text. We have also added reference numbers to the figure legend of figure 2. We agree that this makes it much easier for readers to navigate the literature.

- I think, it may confuse the readers that whereas the PHOT directed processes occur in the membrane and cytosol, are clearly linked to cell growth; the molecular actions happen in the nucleus are not. I feel, that only a few arrows and some sentences explaining how nuclear signalling reaches the cell wall and affects growth are missing.

>>Thank you for this helpful suggestion. As requested, Figure 1 is updated with additional arrows to clarify that nuclear signalling affects cell growth via regulation of cell wall responses, as well as other processes. Upon this question, we realized that our text on the matter was a bit too brief and we expanded it accordingly, from line 372 onwards

Optionally a paragraph could be inserted, discussing shortly what happens in darkness when photoreceptor mediated pathways are inactive explaining how auxin pathways contribute the skotomorphogenic growth. Maybe a modified Fig 1. could demonstrate how auxin signalling is working without photoreceptor action.

>>We appreciate the suggestion of this addition, but suggest that the regulation of auxin in the skotomorphogenesis-inducing dark situation actually follows from the reverse of our explanation of the light situation. This is also true for Figure 1 where darkness would be represented by a lack of nuclear UVR8, cry and phy and general inactivity of phot1. We made a minor change on line 63 to further explain that light activates the photoreceptors.

Reviewer 3 Report

In the manuscript “Photoreceptors regulate plant developmental plasticity through auxin” Jesse J. Küpers, Lisa Oskam and Ronald Pierik review recent advances in the crosstalk between light perception and auxin signaling in adaptive plant development. The review is written in a concise to-the-point way, covers the relevant literature and fits into the scope of the Plants pecial Issue "Advances in Auxin Research". Provided several issues I have with the present form of the manuscript (see below) are resolved, this paper is in my opinion suitable for publication in Plants.

Major comments

Shade Avoidance is the second keyword of the paper, yet in the text the term first appears on line 153 when describing a mutant, without any explanation. This concept is crucial for the paper and should therefore in my opinion be introduced and briefly explained in the general introduction to light perception (section 2, lines 53-63).

On line 304, “Upon nuclear arrival” does not read well, and the whole sentence implies the TIR1/AFB-Aux/IAA pathway is the only mechanism of auxin perception, which is not fully consistent with the literature. Furthermore, the statements that “auxin is perceived by the TRANSPORT INHIBITOR RESISTANT1 (TIR1) and AUXIN SIGNALING F-BOX (AFB) receptors” (lines 304-305) and “Once the SCFTIR1/AFB complex binds auxin, it interacts with and ubiquitinates the auxin signalling repressors AUXIN/INDOLE ACETIC ACID (AUX/IAAs)” (lines 313-314) are somewhat misleading, as the commonly accepted model sees TIR1/AFB and Aux/IAA proteins as auxin co-receptors (see for example Prigge et al., 2020 and many other works).

This section should be reformulated to acknowledge that 1) auxin is perceived by the TIR1/AFB-Aux/IAA co-receptor complex rather than by TIR1/AFBs alone, and 2) this is the main, but not the only mechanism of auxin perception.

In the paragraph on the role of auxin signaling asymmetry and the role of PIN3 therein in phototropism (lines 228-234), the reference to Friml et al., 2002 must be included.

Minor remarks

On lines 215-216, passive influx of auxin into the cells should be mentioned as well.

On line 218, the term “canonical PINs” is used without any explanation – the term should either be defined or removed.

On lines 215 and 217, It should be indicated that ref. 57 is a review, as on line  279, or primary references should be cited instead (for example Petrasek et al., 2006 and Wisniewska et al., 2006).

Author Response

Shade Avoidance is the second keyword of the paper, yet in the text the term first appears on line 153 when describing a mutant, without any explanation. This concept is crucial for the paper and should therefore in my opinion be introduced and briefly explained in the general introduction to light perception (section 2, lines 53-63).

>>We thank the reviewer for their great eye for detail and have introduced the term shade avoidance to line 56-57 where we discuss the growth responses.

On line 304, “Upon nuclear arrival” does not read well, and the whole sentence implies the TIR1/AFB-Aux/IAA pathway is the only mechanism of auxin perception, which is not fully consistent with the literature. Furthermore, the statements that “auxin is perceived by the TRANSPORT INHIBITOR RESISTANT1 (TIR1) and AUXIN SIGNALING F-BOX (AFB) receptors” (lines 304-305) and “Once the SCFTIR1/AFB complex binds auxin, it interacts with and ubiquitinates the auxin signalling repressors AUXIN/INDOLE ACETIC ACID (AUX/IAAs)” (lines 313-314) are somewhat misleading, as the commonly accepted model sees TIR1/AFB and Aux/IAA proteins as auxin co-receptors (see for example Prigge et al., 2020 and many other works).

This section should be reformulated to acknowledge that 1) auxin is perceived by the TIR1/AFB-Aux/IAA co-receptor complex rather than by TIR1/AFBs alone, and 2) this is the main, but not the only mechanism of auxin perception.

>>We have made the suggested adjustments in the text in lines 310 and 312-314

In the paragraph on the role of auxin signaling asymmetry and the role of PIN3 therein in phototropism (lines 228-234), the reference to Friml et al., 2002 must be included.

>>The reference to Friml et al., 2002 was added to line 235.

On lines 215-216, passive influx of auxin into the cells should be mentioned as well.

>>Passive influx of protonated auxin was added to line 219

On line 218, the term “canonical PINs” is used without any explanation – the term should either be defined or removed.

>>The term canonical was removed

On lines 215 and 217, It should be indicated that ref. 57 is a review, as on line  279, or primary references should be cited instead (for example Petrasek et al., 2006 and Wisniewska et al., 2006).

>>Suggested change in citation was incorporated on line 218. Primary references suggested above were added to the manuscript as well in line 221.